# Using Explainable Artificial Intelligence to Discover Interactions in an Ecological Model for Obesity

**DOI:** 10.3390/ijerph19159447

**Published:** 2022-08-02

**Authors:** Ben Allen, Morgan Lane, Elizabeth Anderson Steeves, Hollie Raynor

**Affiliations:** 1Department of Psychology, University of Kansas, 1415 Jayhawk Blvd, Lawrence, KS 66045, USA; 2Department of Psychology, University of Tennessee, Austin Peay Building, Knoxville, TN 37996, USA; mlane38@vols.utk.edu; 3Department of Nutrition, University of Tennessee, 1215 W. Cumberland Ave., Knoxville, TN 37996, USA; eander24@utk.edu (E.A.S.); hraynor@utk.edu (H.R.)

**Keywords:** adolescent obesity, neighborhood education, neighborhood poverty, household income, parent education, explainable artificial intelligence, machine learning, ecological theory

## Abstract

Ecological theories suggest that environmental, social, and individual factors interact to cause obesity. Yet, many analytic techniques, such as multilevel modeling, require manual specification of interacting factors, making them inept in their ability to search for interactions. This paper shows evidence that an explainable artificial intelligence approach, commonly employed in genomics research, can address this problem. The method entails using random intersection trees to decode interactions learned by random forest models. Here, this approach is used to extract interactions between features of a multi-level environment from random forest models of waist-to-height ratios using 11,112 participants from the Adolescent Brain Cognitive Development study. This study shows that methods used to discover interactions between genes can also discover interacting features of the environment that impact obesity. This new approach to modeling ecosystems may help shine a spotlight on combinations of environmental features that are important to obesity, as well as other health outcomes.

## 1. Introduction

A critical barrier in obesity prevention and treatment is determining which individual, social, and environmental factors shape our health [1,2,3]. A person’s behavior and the multiple environments they navigate cause obesity [4]. There is evidence that connects features of the environment with different health outcomes, but how do such features interact [5,6]? The social ecological model has been a useful tool to conceptualize how proximal and distal environmental factors influence individual behavior and health outcomes [7]. Yet, current analytic methods limit the discovery of interactions between and within different levels of the social ecological model [8,9]. The focus of this paper is to explore an artificial intelligence approach to discovering interacting ecosystem factors that support obesogenic behaviors.

The multitude of measurements from proximal (i.e., intrapersonal and interpersonal) and distal (i.e., community, organizational, and policy) levels of the ecological model make it more challenging to gain systems-level insights into interacting components of the ecosystem. As an example, consider how features of an impoverished environment might affect obesity prevalence. Obesity is more common in impoverished environments as measured by proximal (i.e., household income) and distal (i.e., neighborhood income) ecosystem features [10,11]. There is an inevitable redundancy between interpersonal and community environments, in that neighborhoods often include households with similar levels of income, wealth, and education. However, there are potentially meaningful differences between neighboring households, suggesting potential interactions between features of the interpersonal and community ecosystem.

The challenge of choosing which feature interactions to study is a barrier to gaining new insights into the principles of how levels of the ecosystem interact to shape health. The growing number of ways to measure community levels of the ecosystem has made it increasingly difficult to specify how every possible interacting feature of the environment affects a particular health outcome. To illustrate the problem, one can calculate the number of interactions using the formula for combinations without repetition [*n*!/*k*!(*n*−*k*)!], where *n* represents the number of unique features and *k* represents the number of features in the interaction. For example, 627 interactions are possible with 10 features, and 21,679 interactions are possible with 20 features (up to 5-way interactions). The large search space of potential interactions between the growing numbers of features from the ecosystem levels is too extensive to study interactions one at a time.

The immense number of potential interactions and limitations of commonly used analytic approaches often results in obesity studies that only consider fragments of the ecological model [12,13,14]. As noted above, manually selecting less than a handful of interactions is ignoring hundreds or even thousands of other potentially meaningful interactions. Yet, a key element of ecological theories is the interactions between individual behavior and a variety of environmental features [7,15]. A recent scoping review of the childhood obesity literature suggests that most published studies are limited to interactions with a single level of the ecological model (i.e., interpersonal and community, etc.) [16]. To advance our understanding of how ecosystem features interact to cause obesity requires an efficient approach to searching for interactions within and across levels.

A potential solution to the problem of identifying important interactions among multi-feature, multi-level ecosystems is to leverage machine-learning algorithms that can train a computer to learn what complex feature interactions are important for predicting a particular outcome [17]. Traditionally, people use machine-learning models to predict an outcome, not explain feature patterns in the model [18]. This is in part because machine-learning algorithms can produce highly complex solutions that are difficult to interpret [19]. However, recent research in artificial intelligence has helped build interpretable or explainable machine-learning models that show domain knowledge about the modeled features [20,21,22]. Artificial intelligence solves problems using machines to do the tasks. With explainable artificial intelligence, the task is the extraction of knowledge from a machine-learning model to gain insights into phenomena being modeled [17,23,24]. Thus, such an approach could help to extract information on interacting ecosystem features from a machine-learning model used to predict a particular health outcome.

Here, explainable artificial intelligence is used to discover interacting features of a complex multi-level environment that reinforces obesity/obesogenic behaviors. The goal is to use adolescent obesity as a proof-of-concept case for using a machine learning approach to better understand the interactions between components of an ecosystem. Using components from intrapersonal, interpersonal, and community, this paper shows evidence that random forest models can learn interactions between features of the ecosystem that predict obesity in youth. Taking a multi-level view of ecological systems, the models include features from both the proximal (i.e., intrapersonal and interpersonal) and distal (i.e., community) environment. The discovered interactions are discussed along with implications and limitations of the approach for environmental research.

## 2. Materials and Methods

### 2.1. Human Participants

Here we use data collected as part of the ABCD study (https://abcdstudy.org, Release 3.0.1) accessed on 28 July 2021. The data came from 22 sites in the United States from 11,875 human participants aged 9–10 years. Most of our analysis is with the baseline data available as part of the ABCD Study Curated Annual Release 3.0.1 (https://data-archive.nimh.nih.gov/abcd, accessed on 28 July 2021). We also included dietary intake data that were first collected at the 1-year follow-up visit. The primary outcome of our analyses is age and sex-adjusted waist-to-height ratio z-scores, for which there were data on 11,112 participants. We excluded participants from our analyses who were missing waist circumference, height, age, or sex data. The Institutional Review Board at the University of Tennessee approved this project.

### 2.2. Predictive Variable Selection

We started our analyses with 120 features used as predictors of each child’s waist-to-height ratio. We selected these features to provide information about individual characteristics of the child (i.e., pubertal stage, nutrition, and physical activity), their household (i.e., family history and parent demographics), and neighborhood environment (i.e., crime, pollution, and poverty). We began the variable selection process with the goal of taking advantage of the immense ABCD data collection. No single feature was missing over 5% of observations. However, a limitation of the random forest modeling strategy utilized here is that the input data cannot include missing values. Thus, to estimate missing data, we used the multivariate imputation by chained equations R package (M.I.C.E. version 3.13.0) and created separate models for each of the data partitions (described below) [25]. For all datasets, we performed 5 imputations after 40 iterations and used the median imputed value for the final analysis. To check for convergence, we inspected the trace lines of means and standard deviations across iterations for each variable and compared means and standard deviations before and after imputation. We did not use waist-to-height ratio, our primary outcome, to impute any variable. Convergence plots and the R code used to impute the missing data are available in Appendix A.

### 2.3. Data Partitioning

For model training and evaluation, we employed a stratified 2-fold cross-validation partitioning scheme. We created two partitions of the full dataset (1st partition: 5561 participants, 2nd partition: 5551 participants) using the groupdata2 R package (version 1.4.1). We balanced the partitions by our primary outcome (waist-to-height ratio z-scores) and such that siblings stayed in the same set. The R code used to partition the data is available in Appendix A.

### 2.4. Intrapersonal Features

#### 2.4.1. Waist-to-Height Ratio Z-Scores

The primary outcome measure in our analyses was age and sex-adjusted z-scores for waist-to-height ratio. We chose this metric because it adjusts for height, age, and sex differences, and is a better surrogate measure of percent body fat than body mass index [26]. We calculated standardized scores using the childsds R package (version 0.7.6), which is based on the distribution of waist-height ratio in children aged 5–19 years in NHANES III [27]. To perform the calculation, we retrieved waist circumference, height, sex, and age from the ABCD Youth Anthropometrics file named abcd_ant01. To account for extreme and/or unlikely input data for the calculation of waist-to-height ratios, we removed individuals with an age and sex adjusted z-score of ≥4 or ≤−4 using the waist-to-height ratio z-scores and a CDC reference table for height, weight, and body mass index [28]. R code for calculating the age and sex-adjusted z-scores for waist-to-height ratio is available in Appendix A.

#### 2.4.2. Pubertal Stage

Given the age range of the ABCD sample, pubertal stage was included as an individual-level variable. Pubertal stage is based on the research subjects’ sex at birth and parent responses to the Pubertal Development Scale [29]. The scale values range from 1 to 5 (1 = prepuberty; 2 = early puberty, 3 = mid puberty, 4 = late puberty, 5 = post puberty). These data are available in the ABCD Sum Scores Physical Health Parent file named abcd_ssphp01.

#### 2.4.3. Race and Hispanic Ethnicity

We also include race and ethnicity as individual-level variables, given their known relationship with obesity disparities. Ethnicity is a binary parental response (yes/no) to whether they consider their child Hispanic, Latino, or Latina. Parental responses (yes/no) to whether their child belonged to a particular racial group included 16 binary questions (White, Black/African American, American Indian/Native American, Alaska Native, Native Hawaiian, Guamanian, Samoan, Other Pacific Islander, Asian Indian, Chinese, Filipino, Japanese, Korean, Vietnamese, Other Asian, Other Race). These data are available in the ABCD Parent Demographics Survey file named pdem02.

#### 2.4.4. Dietary Information

Questions from a nutritional assessment completed by the parent provided information about the child’s diet in a typical week over the past year, as well as whether the biological mother took prenatal vitamins. Child nutrition information comprised 14 binary responses (yes/no) to questions about a wide range of food groups with a fixed consumption frequency: “Whole grains 3 or more times per day”, “Green leafy vegetables 6 or more times per week”, “Other vegetables 1 or more time per day”, “Berries 2 or more times per week”, “Red meats and meat products less than 4 times per week”, “Fish 1 or more time per week”, “Poultry 2 or more times per week”, “Beans 4 or more times per week”, “Nuts 5 or more times per week”, “Fast food or fried food less than 1 time per week”, “Olive oil is used as the primary oil”, “Butter or margarine is used less than 1 Tablespoon per day”, “Cheese less than 1 time per week”, and “Pastries or sweets less than 5 times per week”. There are also two questions asking whether the biological mother took daily prenatal vitamins or folic acid supplements before or during pregnancy. These data are available in the ABCD Child Nutrition Assessment file named abcd_cna01.

#### 2.4.5. Physical Activity

The ABCD study collected information about physical activity in different contexts. They based three indicators on items from the Youth Risk Behavior Survey. These questions assessed how many of the past 7 days the youth was physically active for at least 60 min per day, how many of the past 7 days the youth did exercises to strengthen or tone your muscles, and how many days in an average school week does the youth go to physical education (PE) class. These data are available in the ABCD Youth Risk Behavior Survey Exercise Physical Activity file named abcd_yrb01.

They also asked parents of children in the ABCD study about their child’s involvement in 23 different sports in the past 12 months. We calculated the average time spent per week in each sport. We summed the values of all the sports to estimate involvement in sports on a weekly time scale. These data are available in the ABCD Parent Sports and Activities Involvement Questionnaire (SAIQ) file named abcd_saiq02.

### 2.5. Interpersonal Features

#### 2.5.1. Developmental History Measures

Items pertaining to the child’s developmental history provided information about the youth’s completion of developmental milestones, medical problems during birth and pregnancy, and prenatal substance exposure. Six initial questions assessed whether they administered the exam to the caregiver in Spanish, whether they were the biological mother, age of biological mother and father at birth, whether the child has a twin, and whether the child was a planned pregnancy. 16 binary responses (yes/no) assessed drug use during pregnancy before and after they knew about being pregnant (prescription medications, tobacco, alcohol, and marijuana. The ABCD study assessed use of other drugs (i.e., cocaine, heroin, OxyContin, and morphine) that we chose not to include because less than 1% of the sample reported affirmative responses. Prenatal vitamin and caffeine consumption during pregnancy was also assessed (yes/no). They assessed 25 different complications with the pregnancy or at birth using binary response variables (yes/no). Additional information included length of time in months being breastfed, age when first rolled over, said first word, and walked without help. They also asked caregivers whether their child’s motor and language development was earlier, average, or later than most other children (1 = Much earlier; 2 = Somewhat earlier; 3 = About average; 4 = Somewhat later; 5 = Much later). Caregivers also reported whether their child had ever wet the bed at night. These data are available in the ABCD Developmental History Questionnaire file named dhx01.

#### 2.5.2. Parent Demographics and Familial Environment

Information collected from the parent as part of a demographic intake form characterized the household environment. These questions assessed the parental marital status, education, work status, earnings of primary caregiver before taxes, whether they have a partner, number of people living in the house, and total combined family income. The remaining questions showed whether anyone in the immediate family had experienced various events in the past 12 months: “Needed food but couldn’t afford to buy it or couldn’t afford to go out to get it”, “Were without telephone service because you could not afford it”, “Didn’t pay the full amount of the rent or mortgage because you could not afford it”, “Were evicted from your home for not paying the rent or mortgage”, “Had services turned off by the gas or electric company, or the oil company wouldn’t deliver oil because payments were not made”, “Had someone who needed to see a doctor or go to the hospital but didn’t go because you could not afford it”, and “Had someone who needed a dentist but couldn’t go because you could not afford it”. These data are available in the ABCD Parent Demographics Survey file named pdem02.

### 2.6. Community Features

The residential address of each participant made it possible to characterize the community environment using information from external geocoded databases. Included was one indicator of crime: number of total crimes. Excluded were more specific indicators of crime (i.e., adult offenses, violent crimes, drug abuse violations, drug sale, marijuana, drug possession, and driving while under the influence) because of higher inter-correlations (r > 0.80) with each other and with total crime. Other measures characterized the neighborhood in terms of deprived education, housing-quality, and poverty. Additional neighborhood environment characteristics included metrics of population density, proximity to major roads (meters), walkability, and ambient air pollutants at 10 × 10 km^2^ [annual average of fine particulate (PM_2.5_) and the three-year average of nitrous di-oxide (NO_2_)]. These data are available in the ABCD Residential history derived scores file named abcd_rhds01.txt.

#### Model Training

The input for the data analysis pipeline is a set of features (i.e., intrapersonal, interpersonal, and community) and the predicted outcome (i.e., waist-to-height ratio z-scores). This input is used by the iterative random forest (iRF) R package (version 3.0.0) to build prediction models of the waist-to-height ratio z-scores using a 2-fold cross-validation scheme [23,30,31]. The algorithm works by first generating a forest of 1000 decision trees using the training data. It generates each decision tree using a subset of 10 features $(\sqrt{120\;features}$), selected at random from the entire set of 120 features. It estimates the importance of each feature based on the average variance explained in the outcome across all the decision trees. After the algorithm generates the first prediction model, it generates a second prediction model using the same process with one exception: the algorithm selects the subset features such that the probability of each feature being selected is weighted based on their performance in the first prediction model. The best performing features in the first prediction model are more likely to be included in the decision tree than poor performing features. The algorithm iterates through this process 5 times and keeps the model from the iteration with the best performance based on out-of-bag error. We built two prediction models following this process using the two data partitions.

The output of the iRF algorithm is a list of stable interactions between variables that predict waist-to-height ratio. The list of interactions is generated by decoding the final prediction model using random intersection trees [32]. The algorithm identifies interactions by detecting co-occurring features that show similar decision rules on the decision paths of the final iteration of the random forest model. Taking the intersection of interactions discovered by the two prediction models implemented a 2-fold cross-validation scheme. Only the interaction term was used to calculate the variance explained in the holdout data for each prediction model. Interactions reported in the results here explained at least 1% of the variance in waist-to-height ratio. The results section shows the median splitting value for the components of each interaction to aid interpretation.

## 3. Results

Table 1 shows descriptive statistics for demographics and waist-to-height ratio z-scores of the whole sample. After randomly partitioning the data into halves, two separate prediction models were built for the waist-to-height ratio z-scores. Each model included 120 features that described individual characteristics of the child (i.e., intrapersonal), their household (i.e., interpersonal), and neighborhood (i.e., community). Using a 2-fold cross-validation scheme, performance was evaluated for each model using the data partition not used in model training. The two models showed modest predictive performance, accounting for a median 11.06% of the variance in the evaluation data (Model_1_ = 10.37%, Model_2_ = 11.75%). Figure 1 shows the average importance of the ten most important features, averaged across both prediction models. All of the most important model features described some aspect of the neighborhood.

### 3.1. Summary of Random Forest Prediction Models

Figure 2 shows the average variance explained for each of the six interactions discovered by the prediction models. Table 2 shows the median threshold values of the decision rules used for each interacting variable. The algorithm found interactions between interpersonal and community factors, intrapersonal and community factors, and between different community factors.

### 3.2. Interpersonal and Community Level Interactions

Children of parents with less formal education had a higher waist-to-height ratio z-score if they lived in a neighborhood with a low percentage of adults with a high school diploma (see Figure 3A). Children living in a home with low total household income had a higher waist-to-height ratio z-score if they lived in a neighborhood with a high percentage of residents living in poverty (see Figure 3B). Children living in a home with low total household income also had a higher waist-to-height ratio z-score if they lived in a neighborhood with lower levels of particle pollution (see Figure 3C).

### 3.3. Intrapersonal and Community Interactions

Children living in a low-income neighborhood had a higher waist-to-height ratio z-score if they played sports for less than 23 min per week (see Figure 3D).

### 3.4. Interactions between Community Factors

The last two discovered interactions involved features within the community level. Children living in a neighborhood with a low percentage of adults with a high school diploma had a higher waist-to-height ratio z-score if they lived in a neighborhood with high median home values (see Figure 3E). Children living in a neighborhood with a high percentage of households below the poverty line had a higher waist-to-height ratio z-score if they lived in a neighborhood with a high number of single-parent households (see Figure 3F).

## 4. Discussion

In this study, an explainable artificial intelligence approach helped discover interactions between obesogenic features of the multiple environments that youth navigate. Using components from intrapersonal, interpersonal, and community, random forest models learned interactions between features of the ecosystem that predict obesity in youth. The main findings are interactions that show compounding obesogenic risks across and within ecosystem levels. The cross-sectional design stifles causal claims between the discovered interactions and childhood obesity. However, this study shows strong evidence of unique obesogenic risk across intrapersonal, interpersonal, and community levels of the ecosystem. This study also shows that the probing of machine learning models is potentially fruitful for the discovery of novel intervention targets in obesity prevention.

The two most potent interactions predictive of high waist-to-height ratios reported here are examples of so-called double-jeopardy effects in that they show compounding risks from different ecosystem levels [33]. Discovery of interactions between economic or educational resources in the household and community is consistent with evidence showing associations between these metrics and obesity. Evidence of interactions between interpersonal and community factors supports the idea that youth with the same household economic and educational resources can have different obesity risk levels depending on the economic and educational resources in their neighborhood [34]. Though the mechanisms are not clear, lower levels of education and economic resources could cause differences in food availability or consumption in the home and neighborhood that promote eating habits that cause obesity [35,36]. Future releases of the ABCD dataset that include features of the neighborhood food environment can make it possible to address these hypotheses.

The predictive models discovered compounding obesogenic risk for youth living in low-income neighborhoods that engage in little to no weekly sports activities. This interaction differs from the previous double-jeopardy effects in that the compounding risks come from different constructs (i.e., economic resources and physical activity). At face value, this interaction suggests an intervention geared towards increasing engagement in sports for youth in low-income communities [37,38,39]. However, the predictive models did not include a measure of physical activity facilities or other community level features that might act as a barrier to sports participation. Perhaps the interaction between neighborhood income and sports activity is a proxy for a subset of low-income communities that also have barriers to sports participation [40,41]. Conceivably sports participation is a proxy for having a supportive adult that facilitates their activities and provides general care. Addressing these hypotheses requires the combination of parenting indices and features of the built environment, such as proximity to public sport facilities and green spaces.

The prediction models also discovered compounding barriers within the community level. Here, the finding was that impoverished neighborhoods with a high proportion of single-parent households are more obesogenic than impoverished neighborhoods with a low proportion of single-parent households. Previous studies show higher obesity rates associated with living in a single-parent household and living in a neighborhood with a high proportion of single-parent households [32,42]. Additionally, the prediction models also included a measure of the parent’s marital status, though it was the proportion of single-parent households in the neighborhood that interacted with neighborhood poverty. Perhaps neighborhoods with high levels of poverty and single-parent homes have a poor food environment compared to impoverished neighborhoods with a higher percentage of two-parent households.

A paradoxical finding in this study was that two of the cross-validated interactions described both high and low levels of two risk factors predictive of higher waist-to-height ratios compared to high levels of both factors. For example, youth living in neighborhoods characterized by low neighborhood education levels and high median home values had higher predicted waist-to-height ratios compared to youth living in neighborhoods characterized by low neighborhood education levels and low median home values. The pattern of findings for neighborhood education is consistent with studies showing higher rates of obesity in adults without a high school degree [43], and studies showing higher rates of obesity in children living in neighborhoods with a higher proportion of adults without a high school degree [44]. However, the finding of higher median home values corresponding to higher obesity risk conflicts with reports showing higher rates of obesity in neighborhoods with lower home values [45,46,47]. Indeed, the ABCD study data used here show an overall main-effect of a negative linear correlation between home values and waist-to-height ratio z-scores. Yet the surface plot for this interaction (see Figure 3F) shows that neighborhoods characterized by low educational resources and high median home values are more obesogenic than neighborhoods characterized by high educational resources and low median home values. If true, this paradoxical effect highlights the potential for the myriad combinations of factors to create different ecosystems of obesogenic risk.

Overall, this study shows evidence that multiple levels of a youth’s ecosystem interact to exacerbate the obesity risk. The findings support the notion that community, interpersonal, and intrapersonal features are proxies for non-overlapping processes that influence the propagation of obesity [32]. The findings also show that ecosystem features can interact in unexpected ways, which bolsters the importance of using artificial intelligence to discover interacting ecosystem features. In doing so, the algorithm revealed subsets of individuals that deviated from the expected linear patterns.

While our data-driven approach and cross-validation scheme have several strengths relative to previous studies of the environment and obesity, reliance on only observational data is a limitation. For example, a longitudinal study of youth who move from low- to high-income neighborhoods would provide more interesting evidence of how neighborhood and household economic resources intersect. However, even with multiple waves of data and naturally occurring interventions, the mechanisms for which these interactions are a proxy need to be defined prior to any public policy change. Yet, a notable limitation to discovering such mechanisms is the common approach to combine different community or interpersonal features into an abstract composite measure that is difficult to interpret relative to the individual features themselves.

A methodological limitation of this study is the use of census-based proxies of geographical boundaries for neighborhoods. It is unclear to what degree these boundaries correspond to the geographical distribution of causal factors linking obesity to features of the community. Multiple economic features contributed to the predictive models, suggesting that single indices of median neighborhood levels may not accurately characterize heterogeneous communities. Neighborhood factors aggregated based on the census tract were the most important features in the predictive models reported here. More broadly, these findings bolster the importance of the community ecosystem as a determinant of obesity not accounted for by intrapersonal factors [6].

## 5. Conclusions

In summary, this paper shows an explainable artificial intelligence approach to searching for interactions between ecosystem features that predict obesity and other health outcomes. In practice, the method will allow a more comprehensive analysis of health and environment interactions that is better aligned with the theoretical framework of ecological theories. This expanded approach to searching for interactions has the potential to improve knowledge of how features of the environment interact within and across levels of ecological models for health. A more comprehensive knowledge of these interactions is likely to inform social programs aimed at preventing obesity in youth. Here, many of the interactions involve either economic or educational resources highlighting their importance for directing interventions. Open questions remain about the operating mechanisms at play at the community level that interact with individual behavior and propagate obesity risk.

## Figures and Tables

**Figure 1 ijerph-19-09447-f001:**
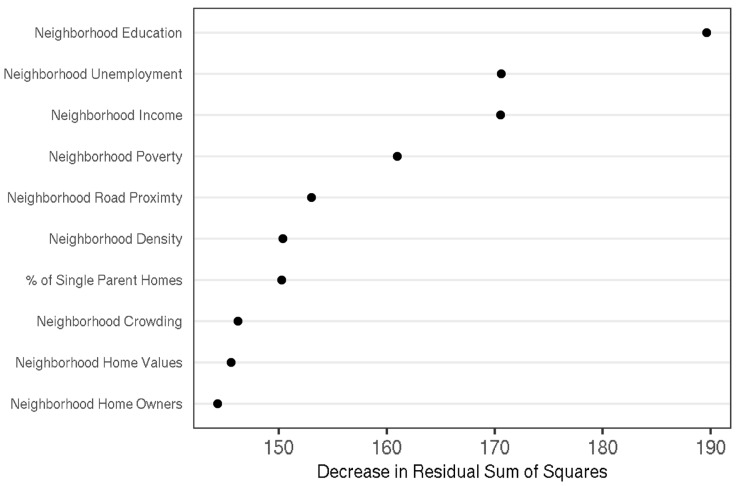
Ten most important features in predicting waist-to-height ratio z-scores. Dotplot showing the decrease in residual sum of squares when the decision trees include each of the most important features. Importance values are averages of the two prediction models. The most important features all characterize some aspect of the neighborhood.

**Figure 2 ijerph-19-09447-f002:**
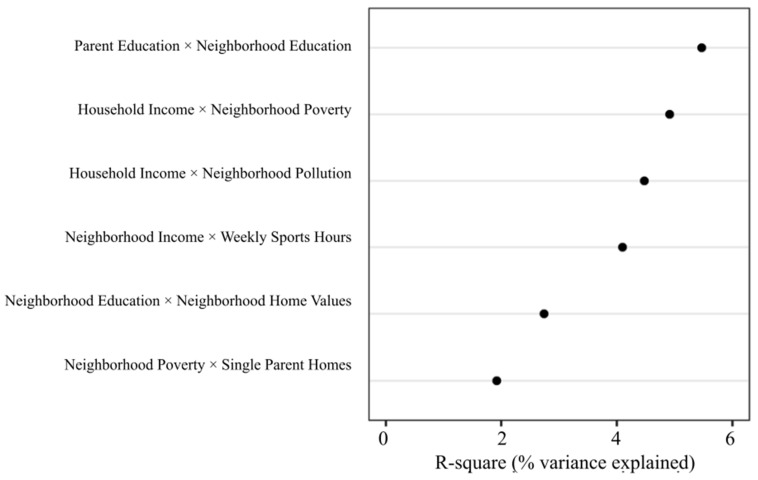
Cross-validated interactions predictive of waist-to-height ratio z-scores. Evaluation of the interactions was based on how much variance the interaction explained in the holdout data based on the coefficient of determination (R-square). R-square values presented above are the average of the two predictive models.

**Figure 3 ijerph-19-09447-f003:**
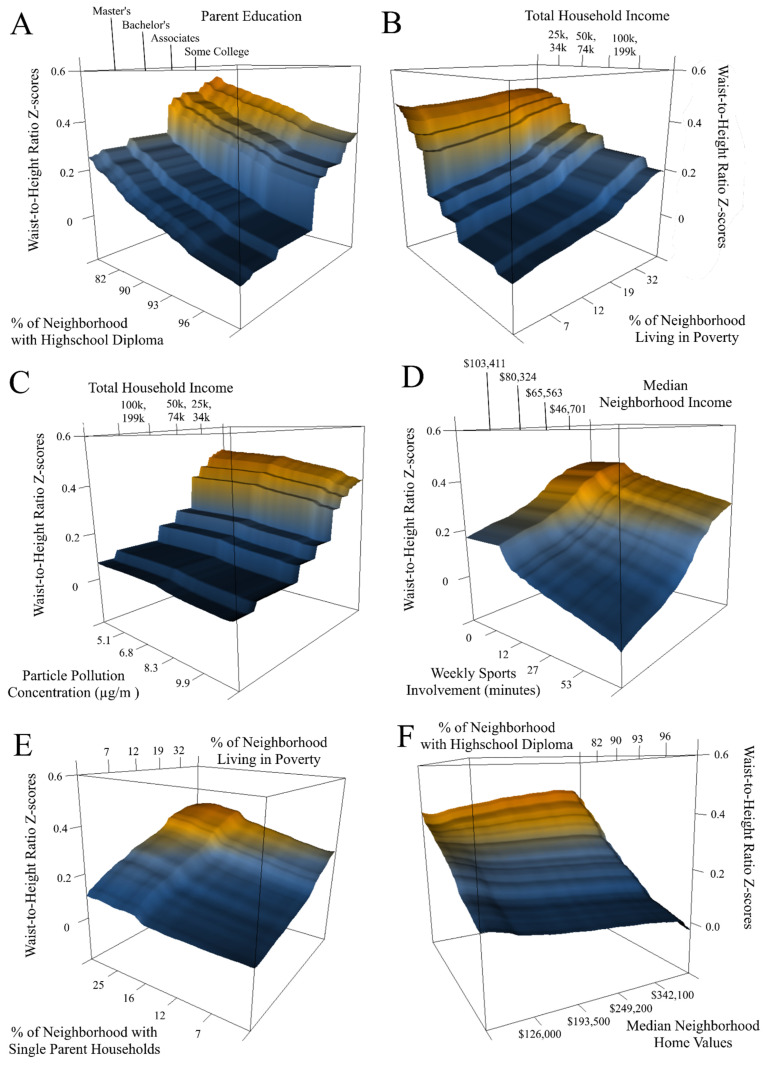
**Feature interactions predicted waist-to-height ratio z-scores.** 3d surface map showing waist-to-height ratio z-scores by quantiles of interacting features. The predicted waist-to-height ratio z-scores as a function of parent education level and % of neighborhood with a high school diploma (**A**), total household income and % of neighborhood living in poverty (**B**), total household income and particle pollution (**C**), median neighborhood income and weekly involvement in sports (**D**), % of single parent homes in neighborhood and % of neighborhood living in poverty (**E**), % of neighborhood with a high school diploma and median neighborhood home values (**F**). All surface maps are drawn using the first prediction model and holdout data to show the generalizability of each interaction.

**Table 1 ijerph-19-09447-t001:** Participant demographics and Waist-to-Height Ratio Z-score (N = 11,112).

**Sex**	
Male, n (%)	5811 (52.3%)
Female, n (%)	5301 (47.7%)
**Age (years)**	
Mean (SD)	9.92 (0.626)
Median [Min, Max]	9.92 [8.92, 11.1]
**Waist-to-Height Ratio (Z-score)**	
Mean (SD)	0.208 (1.05)
Median [Min, Max]	0.215 [−3.99, 3.93]

**Table 2 ijerph-19-09447-t002:** Decision Rules and Median Splitting Values for each interacting variable.

Feature A	Feature B
Parent Education < Bachelor’s Degree	<92% of Neighborhood with High School Degree
Household Income < $50,000	≥18% of Neighborhood Living in Poverty
Household Income < $50,000	Neighborhood Small Particle Pollution <7.9 µg/m^3^
Median Neighborhood Income < $72,341	<23 min of weekly sports
<92% of Neighborhood with High School Degree	Median home values ≥ $215,825
≥18% of Neighborhood below poverty line	<16% Single-parent homes

## Data Availability

Prior to publication, the manuscript will include a NIMH Data Archive Digital Object Identifier for the specific data used in preparing this article.

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
