# Peer review of "Using Explainable Artificial Intelligence to Discover Interactions in an Ecological Model for Obesity"

_ijerph, 2022, doi:10.3390/ijerph19159447_

Round 1
Reviewer 1 Report
Abstract
I suggest using another word instead of “promote”.
“Yet, current multilevel modeling methods are inept in their ability to search for 10
interactions.”
You should mention some multilevel modeling methods that you refer.
“Here, an explainable artificial intelligence approach…...”
You must mention the name of the method. It is known or it is proposed in this paper?
“This paper shows the extraction of interactions between obesogenic features of a complex multi-level environment from machine learning prediction models of waist-to-height ratios. The approach discovered interactions of compounding obesogenic risk levels between and within ecosystem levels.”
Must reformulated to be better understandable.
“This study shows a new approach to teasing apart factors from ecosystem levels that influence obesity the most, which may offer insight into selecting the impactful targets when designing multilevel intervention studies.”
It is not described very clearly a specificity of the proposal. It is not very clear in what consist of the novelty.
The paper is focused on obesity, and it uses a very large dataset. Does it give any added value to the medical domain regarding the obesity?
Keywords: obesity; neighborhood; socioeconomic; multilevel; ecological
Extended these keywords in order to be more expressive. For instance, just to mention one of them “neighborhood” what kind of “neighborhood” refer.
“As an illustration, 627 interactions are possible in an environment with 10 features, 21,679 interactions are possible with 20 features (up to 5- 48 way interactions).”
You must explain how you calculated the number of interactions.
At the end of introduction section, must be included a paragraph that briefly presents the upcoming structure of the paper.
Section 2.
“Participants”
Must be extended in order to illustrate what kind of participants are referred.
“To estimate missing data, we used the multivariate imputation by chained equations R package (M.I.C.E. version 3.13.0) and created separate models for each of the data partitions (described below).[16]”
If the data is missing why is necessary, the estimation? It must be explained more clearly.
Page 3
“(n = 5,561, n = 5,551)”
Revise. Must be used different letters.
Page 3, line 119
“Puberty”
Must be extended to illustrate its significance.
Same for “Race and Ethnicity”, “Nutrition”…..
Page 5, beginning with the line 212
“Model training”
Must be described in algorithmic form, indicating the input, output.
Page 7.
It should be more clearly explained what kind of “R” is calculated.
Author Response
Reviewer 1 - Comments and Suggestions for Authors
Abstract
I suggest using another word instead of “promote”.
# we have changed “promote” to “cause”
“Yet, current multilevel modeling methods are inept in their ability to search for interactions.”
You should mention some multilevel modeling methods that you refer.
# We have revised this sentence to describe the specific aspect of current analytic approaches that is problematic.
“Yet, many analytic techniques, such as multilevel modeling, require manual specification of interacting factors, making them inept in their ability to search for interactions.”
“Here, an explainable artificial intelligence approach…...”
You must mention the name of the method. It is known or it is proposed in this paper?
# This section of the abstract now mentions the specific name and origin of the method.
“This paper shows evidence that an explainable artificial intelligence approach, commonly employed in genomics research, can address this problem. The method entails using random intersection trees to decode interactions learned by random forest models.”
“This paper shows the extraction of interactions between obesogenic features of a complex multi-level environment from machine learning prediction models of waist-to-height ratios. The approach discovered interactions of compounding obesogenic risk levels between and within ecosystem levels.”
Must reformulated to be better understandable.
# We appreciate the encouragement to improve the clarity of the abstract. This section now reads as follows:
“Here, this approach is used to discover interactions between features of a multi-level environment from random forest models of waist-to-height ratios using 11,112 participants from the Adolescent Brain Cognitive Development study. The method helped to discover interactions between and within ecosystem levels. “
“This study shows a new approach to teasing apart factors from ecosystem levels that influence obesity the most, which may offer insight into selecting the impactful targets when designing multilevel intervention studies.”
It is not described very clearly a specificity of the proposal. It is not very clear in what consist of the novelty.
# This portion of the abstract has been revised for clarity and to better highlight the novelty.
“This study shows that methods used to discover interactions between genes can also discover interacting features of the environment that impact obesity. This new approach to modeling ecosystems may help shine a spotlight on combinations of environmental features that are important to obesity, as well as other health outcomes.”
The paper is focused on obesity, and it uses a very large dataset. Does it give any added value to the medical domain regarding the obesity?
# Though not the intent of the paper, there is potential for added value to the medical domain regarding obesity. One example is that all the top ten predictors of obesity came from the neighborhood environment, rather than individual level predictors (i.e., pubertal status, physical activity, nutrition). Neighborhood factors also dominated the discovered interactions. In short, the paper highlights the importance of the environment, where the child lives, besides the traditional focus on individual-level factors.
Keywords: obesity; neighborhood; socioeconomic; multilevel; ecological
Extend these keywords in order to be more expressive. For instance, just to mention one of them “neighborhood” what kind of “neighborhood” refer.
# We extended the keywords to make them more expressive:
Keywords: adolescent obesity; neighborhood education; neighborhood poverty; household income; parent education; explainable artificial intelligence; machine learning; ecological theory
“As an illustration, 627 interactions are possible in an environment with 10 features, 21,679 interactions are possible with 20 features (up to 5- 48 way interactions).”
You must explain how you calculated the number of interactions.
# We now include the formula used to calculate the possible combinations:
To illustrate the problem, calculate the number of interactions using the formula for combinations without repetition [n! / k!(n-k)!], where n represents the number of unique features and k represents the number of features in the interaction. For example, 627 interactions are possible with 10 features, 21,679 interactions are possible with 20 features (up to 5-way interactions).
At the end of introduction section, must be included a paragraph that briefly presents the upcoming structure of the paper.
# The last paragraph has been revised to highlight the upcoming structure of the paper:
“Here, explainable artificial intelligence is used to discover interacting features of a complex multi-level environment that reinforces obesity/obesogenic behaviors. The goal is to use childhood obesity as a proof-of-concept case for using a machine learning approach to better understand the interactions between components of an ecosystem. Using components from intrapersonal, interpersonal, and community, the paper shows evidence that random forest models can learn interactions between features of the ecosystem that predict obesity in youth. Taking a multi-level view of ecological systems, the models include features from both the immediate (i.e., intrapersonal, interpersonal) and distal (i.e., community) environment. The discovered interactions are discussed along with implications and limitations of the approach for environmental research.”
Section 2.
“Participants”
Must be extended in order to illustrate what kind of participants are referred.
# We now note that the participants are “Human Participants”
“To estimate missing data, we used the multivariate imputation by chained equations R package (M.I.C.E. version 3.13.0) and created separate models for each of the data partitions (described below).[16]”
If the data is missing why is necessary, the estimation? It must be explained more clearly.
# Estimation of missing data is necessary because the random forest modeling requires complete data and will not run with missing values. Though the amount of missing data was small (<5%), it was scattered across subjects and variables. Using only complete data would drastically reduce the sample size and waste information. This is now explained in the missing data section:
“However, a limitation of the random forest modeling strategy utilized here is that the input data cannot include missing values.”
Page 3
“(n = 5,561, n = 5,551)”
Revise. Must be used different letters.
# This has been revised as follows:
“We created two partitions of the full dataset (1st partition: 5,561 participants, 2nd partition: 5,551 participants)...”
Page 3, line 119
“Puberty”
Must be extended to illustrate its significance.
# We have extended this header to read “Pubertal Stage” to better reflect the measure. The significance is detailed in the paragraph that follows.
Same for “Race and Ethnicity”, “Nutrition”…..
# We have extended the Race and Ethnicity header to include “Hispanic Ethnicity” to better reflect the measure. The significance is explained in the first sentence of the following paragraph. Similarly, “Nutrition” has been replaced with “Dietary Information”
Page 5, beginning with the line 212
“Model training”
Must be described in algorithmic form, indicating the input, output.
# The input and output of the algorithm are now defined at the beginning of the two model training paragraphs:
“The input for the data analysis pipeline is a set of features (i.e., intrapersonal, interpersonal, and community) and the predicted outcome (i.e., waist-to-height ratio z-scores).”
“The output of the iRF algorithm is a list of stable interactions between variables that predict waist-to-height ratio.”
Page 7.
It should be more clearly explained what kind of “R” is calculated.
# We now note in the figure caption on p. 7 that R2 is the coefficient of determination.
Reviewer 2 Report
Dear authors,
I enjoyed reading your article about using artificial intelligence to discover interactions in an ecological model for obesity. This is a very up-to-date and important topic. Nevertheless, I propose to introduce the following improvements in the manuscript:
1/ In the abstract, it is worth mentioning more about the method that was used.
2/ The author/s indicate that "Traditionally, people use machine learning models to predict an outcome, not explain feature patterns in the model. This is in part because machine learning algorithms can produce highly complex solutions that are difficult to interpret. in artificial intelligence has helped build interpretable or explainable machine learning models that show domain knowledge about the modeled features. [13]" However, they only indicate one publication to support this claim. Please indicate more sources.
3/ The research gap should be described in more detail. In the introduction, it should be indicated why the authors took up this topic and why it is important in the context of theory and practice. This is currently missing.
4/ In the discussion section or conclusions, please indicate the contribution to the theory as well as the practical and social implications of the study.
5/ The article should be checked for linguistic and editorial correctness.
Author Response
Dear authors,
I enjoyed reading your article about using artificial intelligence to discover interactions in an ecological model for obesity. This is a very up-to-date and important topic. Nevertheless, I propose to introduce the following improvements in the manuscript:
1/ In the abstract, it is worth mentioning more about the method that was used.
# The first reviewer made a similar suggestion and we have revised the abstract accordingly.
2/ The author/s indicate that "Traditionally, people use machine learning models to predict an outcome, not explain feature patterns in the model. This is in part because machine learning algorithms can produce highly complex solutions that are difficult to interpret. in artificial intelligence has helped build interpretable or explainable machine learning models that show domain knowledge about the modeled features. [13]" However, they only indicate one publication to support this claim. Please indicate more sources.
# We have cited more sources to support the highlighted statements. The sources provide a historical context for the transition from predicting outcomes using black-box models to using methods to extract meaning and explanations. We also include an empirical study that provides an example of extracting domain knowledge from machine learning models.
Traditionally, people use machine learning models to predict an outcome, not explain feature patterns in the model.
Rai, A. (2020). Explainable AI: From black box to glass box. Journal of the Academy of Marketing Science, 48(1), 137-141.
This is in part because machine learning algorithms can produce highly complex solutions that are difficult to interpret.
Lipton, Z. C. (2018). The mythos of model interpretability: In machine learning, the concept of interpretability is both important and slippery. Queue, 16(3), 31-57.
However, recent research in artificial intelligence has helped build interpretable or explainable machine learning models that show domain knowledge about the modeled features.
Gunning, D., Vorm, E., Wang, J. Y., & Turek, M. (2021). DARPA's explainable AI (XAI) program: A retrospective. Applied AI Letters, 2(4), e61.
Carrieri, A. P., Haiminen, N., Maudsley-Barton, S., Gardiner, L. J., Murphy, B., Mayes, A. E., ... & Pyzer-Knapp, E. O. (2021). Explainable AI reveals changes in skin microbiome composition linked to phenotypic differences. Scientific reports, 11(1), 1-18.
3/ The research gap should be described in more detail. In the introduction, it should be indicated why the authors took up this topic and why it is important in the context of theory and practice. This is currently missing.
# The knowledge gap is now better described in the introduction. We add the following paragraph to show that analytic limitations have created a gap in knowledge about how elements of the environment influence obesity:
“The immense number of potential interactions and limitations of commonly used analytic approaches often results in obesity studies that only consider fragments of the ecological model.12–14 As noted above, manually selecting less than a handful of interactions is ignoring hundreds or even thousands of other potentially meaningful interactions. Yet, a key element of ecological theories is the interactions between individual behavior and a variety of environmental features.7,15 A recent scoping review of the childhood obesity literature suggests most published studies are limited to interactions with a single level of the ecological model (i.e., interpersonal, community, etc.).16 To advance our understanding of how ecosystem features interact to cause obesity requires an efficient approach to searching for interactions within and across levels.”
4/ In the discussion section or conclusions, please indicate the contribution to the theory as well as the practical and social implications of the study.
# The conclusion has been revised to more explicitly talk about the practical, social and theoretical contributions of the paper.
“In summary, this paper shows an explainable artificial intelligence approach to searching for interactions between ecosystem features that predict obesity and other health outcomes. In practice, the method will allow a more comprehensive analysis of health and environment interactions that is better aligned with the theoretical framework of ecological theories. This expanded approach to searching for interactions has the potential to improve knowledge of how features of the environment interact within and across levels of ecological models for health. A more comprehensive knowledge of these interactions is likely to inform social programs aimed at preventing obesity in youth. Here, many of the interactions involve either economic or educational resources highlighting their importance for directing interventions. Open questions remain about the operating mechanisms at play at the community level that interact with individual behavior and propagate obesity risk.”
5/ The article should be checked for linguistic and editorial correctness.
# The article has been thoroughly checked for correctness.
Round 2
Reviewer 1 Report
The paper have been revised according to the revision requests.